# Broadly directed SARS-CoV-2-specific CD4+ T cell response includes frequently detected peptide specificities within the membrane and nucleoprotein in patients with acute and resolved COVID-19

Janna Heide[1,2¤], Sophia Schulte[1], Matin Kohsar[1], Thomas Theo Brehm[1,2], Marissa Herrmann[1,2], Hendrik Karsten[1], Matthias Marget[3], Sven Peine[3], Alexandra M. Johansson[4], Alessandro Sette[5], Marc Lütgehetmann[6], William W. Kwok[4], John Sidney[7], Julian Schulze zur Wiesch[1,2]*

1 Infectious Diseases Unit, I. Department of Medicine, University Medical Center, Hamburg-Eppendorf, Germany, 2 German Center for Infection Research (DZIF), Partner Site, Hamburg-Lübeck-Borstel-Riems, Germany, 3 Department of Transfusion Medicine, University Medical Center, Hamburg-Eppendorf, Germany, 4 Benaroya Research Institute at Virginia Mason, Seattle, Washington, United States of America, 5 Center for Infectious Disease and Vaccine Research, La Jolla Institute for Immunology (LJI), La Jolla, California, United States of America, 6 Institute of Medical Microbiology, Virology and Hygiene, University Medical Center, Hamburg-Eppendorf, Germany, 7 Department of Medicine, Division of Infectious Diseases and Global Public Health, University of California, San Diego (UCSD), La Jolla, California, United States of America

¤ Current address: Department of Obstetrics and Gynecology, University of Chicago, Chicago, Illinois, United States of America
* julianszw@gmail.com

**Data Availability Statement:** All relevant data are within the manuscript and its Supporting

## Abstract

The aim of this study was to define the breadth and specificity of dominant SARS-CoV-2-specific T cell epitopes using a comprehensive set of 135 overlapping 15-mer peptides covering the SARS-CoV-2 envelope (E), membrane (M) and nucleoprotein (N) in a cohort of 34 individuals with acute (n = 10) and resolved (n = 24) COVID-19. Following short-term virus-specific *in vitro* cultivation, the single peptide-specific CD4+ T cell response of each patient was screened using enzyme linked immuno spot assay (ELISpot) and confirmed by single-peptide intracellular cytokine staining (ICS) for interferon-γ (IFN-γ) production. 97% (n = 33) of patients elicited one or more N, M or E-specific CD4+ T cell responses and each patient targeted on average 21.7 (range 0–79) peptide specificities. Overall, we identified 10 N, M or E-specific peptides that showed a response frequency of more than 36% and five of them showed high binding affinity to multiple HLA class II binders in subsequent *in vitro* HLA binding assays. Three peptides elicited CD4+ T cell responses in more than 55% of all patients, namely Mem_P30 (aa146-160), Mem_P36 (aa176-190), both located within the M protein, and Ncl_P18 (aa86-100) located within the N protein. These peptides were further defined in terms of length and HLA restriction. Based on this epitope and restriction data we developed a novel DRB*11 tetramer (Mem_aa145-164) and examined the *ex vivo* phenotype of SARS-CoV-2-specific CD4+ T cells in one patient. This detailed characterization of single T

Information files. The amino acid sequence can be accessed in GenBank under the accession number MT318827.

**Funding:** This project was funded by Deutsches Zentrum für Infektionsforschung (DZIF) (JH, JSzW). JSzW was additionally funded by the DFG (SFB 841 and SFB1328). The project was partially funded by the NIH NIAID under awards AI 142742 (Cooperative Centers for Human Immunology) (AS), National Institutes of Health contract Nr. 75N9301900065 (AS), and U19 AI118626 (AS). WWK and AMJ are supported by NIH grant 3U19AI135817-04S1. The funders had no role in study design, data collection and analysis, decision to publish, or preparation of the manuscript.

**Competing interests:** The authors have declared that no competing interests exist.

cell peptide responses demonstrates that SARS-CoV-2 infection universally primes a broad T cell response directed against multiple specificities located within the N, M and E structural protein.

## Author summary

The SARS-CoV-2 genome encodes for 25 different viral proteins. However, many immunological studies have focused on the immune response against the spike protein. This current study was designed to get a detailed understanding of the breadth and specificity of the CD4+ T cell response directed against the other structural proteins, namely the envelope (E), membrane (M) and nucleoprotein (N) using a comprehensive overlapping peptide set in a cohort of patients during early and resolved COVID-19. We detected a universally broad T cell response with on average more than 20 peptide responses per patient. Three peptides elicited CD4+ T cell responses in more than 55% of all patients, two located within the M protein, and one located within the N protein. These peptides were further defined in terms of length and HLA restriction, and we developed a novel MHC class II tetramer based on this data, which enabled us to investigate the *ex vivo* phenotype of SARS-CoV-2-specific CD4+ T cells in one patient.

This large immunological data set on individual immune responses will be useful for further detailed studies on the immunopathogenesis of SARS-CoV-2 infection and vaccine design.

## Introduction

Coronavirus disease 2019 (COVID-19) caused by the severe acute respiratory syndrome coronavirus type 2 (SARS-CoV-2) is a severe flu-like illness which is associated with hyperinflammation and immune dysfunction. SARS-CoV-2 has led to a pandemic with more than 200 million confirmed cases and more than 4 million deaths (https://covid19.who.int/). Only a small percentage of patients with COVID-19 develop a severe disease course, with the main established risk factors being old age and comorbidities like hypertonus, adiposity or diabetes [1].

SARS-CoV-2 has a single-stranded RNA genome of approximately 30 kb which includes four structural proteins and open reading frames (ORFs) encoding for the nonstructural polyproteins [2]. The structural proteins are the spike (S), envelope (E), membrane (M) and nucleoprotein (N) [3]. The S protein is 1273 amino acids, the M protein 222 amino acids, and the E protein 75 amino acids long. Together, these three antigens are part of the viral coat. The M protein is the most abundant structural protein and it defines the shape of the viral envelope [4]. It has a small N-terminal glycosylated ectodomain, three transmembrane domains, and a much larger C-terminal endodomain that extends 6–8 nm into the viral particle [4]. The 419 amino acids long N protein is involved in the packaging of the RNA genome [5]. The E protein is the smallest of the four structural proteins. This transmembrane protein has an N-terminal ectodomain and a C-terminal endodomain with ion channel activity, which is associated with pathogenesis [6].

Previous studies suggest that SARS-CoV-2-specific T cells play a key role in COVID-19 disease resolution and modulation of disease severity [7–11]. The relationship between T cell immunity against the different SARS-CoV-2 antigens (or pre-existing cross-reactive immune

responses against other coronaviruses) and the clinical course of a SARS-CoV-2 infection are currently being unraveled [7, 9, 12–16]. In case of the original SARS-CoV, most T cell responses were directed against the structural proteins as compared to the non-structural proteins [17]. Furthermore, the T cell responses directed against the M and N protein were among the most dominant and long-lasting [17].

How the adaptive immune response, and in particular T cell response pattern, influence the kinetics of viral loads and the duration of a SARS-CoV-2 infection remains unclear [3, 11]. Additionally, the longevity of the naturally acquired immune memory to SARS-CoV-2 in comparison to the SARS-CoV-2 vaccine-induced response will be a substantial question to address in future studies [11]. The definition of SARS-CoV-2-specific T cell epitopes is important to evaluate potential influences of mutations on acquired immunity and vaccine efficacy. The role of potential cross-reactivity between SARS-CoV-2 and other coronaviruses like SARS-CoV and common cold coronaviruses can also be investigated with the knowledge of established immune epitopes [15, 16]. In this study, we screened a cohort of patients with resolved or acute COVID-19 using a comprehensive, overlapping panel of synthetic 15-mer peptides derived from the SARS-CoV-2 N, M and E protein sequences, rather than relying on preselected and *in silico* predicted epitope specificities [7, 9]. We found a universally broad N, M and E-specific CD4+ T cell response regardless of the clinical course of the disease. The large SARS-CoV-2 epitope data set presented here will be a useful tool for further investigations of the *ex vivo* phenotype of SARS-CoV-2-specific T cells e.g., by tetramer technology, and could also be helpful for efficient peptide-based vaccine design.

## Results

### Clinical features of the study cohort

The clinical data of the patient cohort are summarized in **Tables 1** and **S1**. The cohort consisted of 34 patients infected with SARS-CoV-2. Infection was verified by a positive RT-PCR of nasopharyngeal swabs as previously described [18]. For further analysis, the patient cohort was stratified into patients with acute and resolved COVID-19. PBMC samples of ten patients during acute infection [average days since diagnosis 6.3 (range: 2–16)] and of 24 patients after convalescence [average days since diagnosis 96.36 (range: 40–184)] were collected. 12 patients were female (35%) and 22 were male (65%), the average age was 47.2 years (range: 25–78 years). All patients were treated as inpatients for SARS-CoV-2 infection or attended an outpatient clinic at University Medical Center Hamburg-Eppendorf: 26 patients (76%) were from Germany, eight patients (24%) from the Philippines, Afghanistan, Croatia, Greece, or Syria. 22 patients (65%) had a mild or moderate course of disease as defined by the WHO, while 12 patients (35%) suffered from a severe or critical infection [19]. Seven patients needed to be treated in the intensive care unit during their hospitalization (21%). Five patients were on long-term treatment with immunosuppressive medication due to comorbidities: patients aCov-04, aCov-07 and rCov-03 were treated with mycophenolic acid, doxorubicin/ifosfamide and imatinib, respectively (**S1 Table**) and rCov-12 and rCov-16 received B-cell depleting medication (rituximab and obinutuzumab, respectively).

### High response rate and broadly directed N- and M-specific T cell responses

Using ELISpot as previously described [20–22], we first looked at the *ex vivo* N, M and E-specific IFN-γ production of PBMC from COVID-19 patients upon stimulation with single peptides. A set of 43 overlapping 15-mer peptides covering the M protein, 82 peptides covering the N protein and ten peptides covering the E protein of SARS-CoV-2, were utilized (**S2 Table**). As described earlier [23] and depicted in **S1 Fig,** the *ex vivo* ELISpot after stimulation

**Table 1. Clinical and immunological patient characteristics of acute and resolved COVID-19 patients.** Data are expressed as absolute numbers n (n/N) or n (range) or mean with standard deviation.

| | [normal range] | Acute COVID-19 (n = 10) | Resolved COVID-19 (n = 24) |
|---|---|---|---|
| **Age in years** | | 50.8 (25–76) | 46.58 (26–78) |
| **Sex** | | | |
| Female | | 1 (10%) | 11 (46%) |
| Male | | 9 (90%) | 13 (54%) |
| **Disease severity** | | | |
| Mild | | 5 (50%) | 10 (42%) |
| Moderate | | 2 (20%) | 5 (21%) |
| Severe | | 2 (20%) | 5 (21%) |
| Critical | | 1 (10%) | 4 (17%) |
| **Days since diagnosis** | | 6.3 (2–16) | 96.36 (40–184) |
| **Days since start of symptoms** | | 9.88 (2–24) | |
| **Comorbidities** | | | |
| None | | 3 (30%) | 12 (50%) |
| Hypertension | | 1 (10%) | 7 (29%) |
| Heart disease | | 2 (20%) | 2 (8%) |
| Diabetes | | 3 (30%) | 4 (17%) |
| Lung disease | | 1 (10%) | 2 (8%) |
| Cancer* | | 2 (20%) | 3 (13%) |
| Other | | 3 (30%) | 9 (38%) |
| **Blood cell count at time of analysis** | | | |
| White blood cell count | 3.8–11.0 Mrd/l | 6.98 (± 2.78) | |
| Lymphocyte count | 1.0–3.6 Mrd/l | 1.15 (± 0.53) | |
| Hemoglobin | 14.0–17.5 g/dl | 14.05 (± 4.18) | |
| Platelet count | 150–400 Mrd/l | 226.5 (± 87.84) | |
| **Immunology at time of analysis** | | | |
| T lymphocyte count | 900–2,900/µl | 962.9 (± 550.7) | |
| T lymphocyte % | 55–84 % | 68.3 (± 8.9) | |
| CD4 count | 500–1,350/µl | 587.9 (± 306.7) | |
| CD4 % | 31–60 % | 42.8 (± 8.4) | |
| CD8 count | 290–930/µl | 327.4 (± 250.3) | |
| CD8 % | 13–41 % | 22 (± 6) | |
| CD4/CD8 ratio | 0.6–3.6 | 2.1 (± 0.8) | |
| Tregs % | 5.7–10.1 % | 7.5 (± 2.0) | |
| **Clinical parameters at time of analysis** | | | |
| CRP | -5 mg/l | 89.63 (± 81.54) | |
| IL-6 | <7 ng/l | 61.58 (± 86.51) | |
| Ferritin | 22–322 µg/l | 1066.03 (±1013.16) | |
| Procalcitonin | -0.5 µg/l | 0.39 (± 0.68) | |
| D-dimer | 0.21–0.52 mg/l | 0.69 (±0.49) | |

*aCov-04 received Mycophenolic acid

aCov-07 received Doxorubicin/Ifosfamide and r-Cov-03 received Imatinib

rCov-12 received Rituximab and rCov-16 received Obinutuzumab

with single peptides showed a low overall IFN-γ response with a magnitude barely above the limit of detection of this assay (number of spots range: 0–9 spots per 100,000 cells). We detected IFN-γ responses in all three patients with an average number of nine M peptide

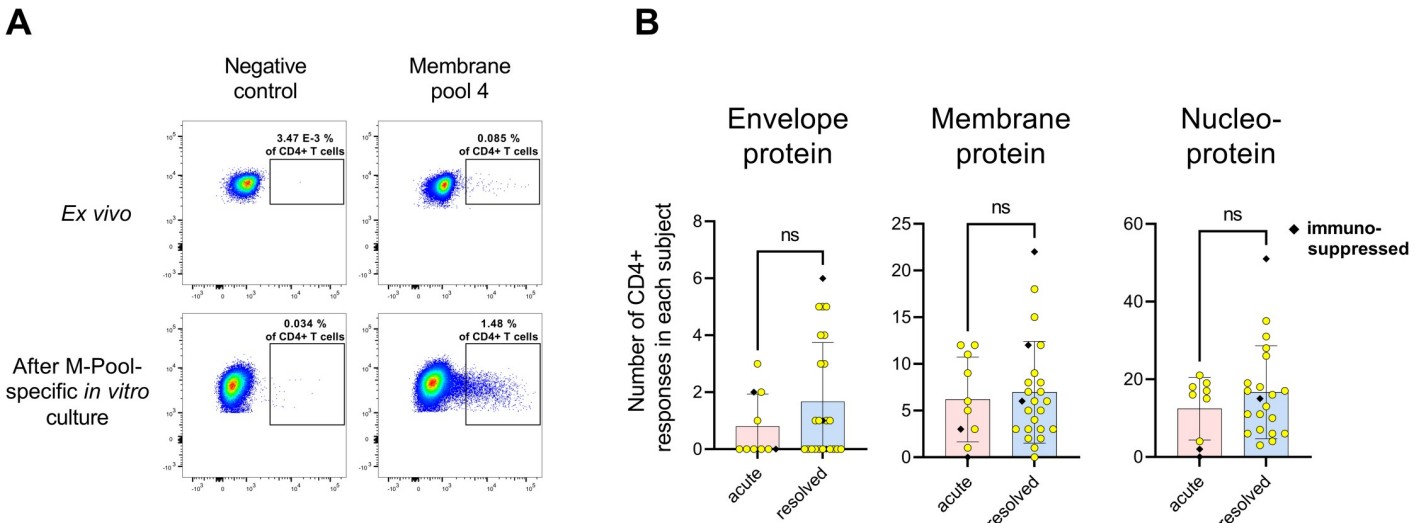

**Fig 1. The envelope, membrane and nucleoprotein-specific CD4+ T cell response. (A)** Comparison of the IFN-γ response after stimulation with M-Pool 4 *ex vivo* and after ten-day T cell culture with M-Pool 4 (patient rCov-01). Gated on CD4+ T cells. R10 and DMSO were added to the negative control. **(B)** Comparison of the total number of envelope, membrane and nucleoprotein-specific CD4+ T cell responses of each patient after 10-day *in vitro* culture. Mean with standard deviation. NS = not significant.

responses per patient (range 5–15 peptide responses), and an average of 15.6 peptide responses (range 9–24 peptide responses) against the N protein.

In order to increase the overall detection rate of individual peptide responses, we employed a well-established highly sensitive *in vitro* approach to detect virus-specific T cell responses [20, 22]. In short, our peptide set was divided into 13 different peptide pools (**S2 Table**), and 13 individual cell cultures were started in parallel per patient with each peptide pool for ten days, followed by a single peptide ELISpot for IFN-γ-production. Each positive T cell response in the ELISpot assay was confirmed in a subsequent ICS assay for IFN-γ production after re-stimulation with the respective single peptide. **Fig 1A** shows an exemplary ICS result of an M peptide pool specific CD4+ T cell response *ex vivo* and after expansion of antigen-specific T cells with M-Pool 4 for ten days.

With this approach, we were able to detect a broad range of IFN-γ CD4+ T cell responses against the M, N and E protein with the majority of responses directed against the N protein (**Fig 1B**). 97% of all patients showed a peptide response against the N protein (29/30) (**Fig 2**). In the resolved patient group, all patients showed at least three or more peptide responses with a mean of 16.67 (range 3–51; median 6) peptide specific CD4+ T cell responses per patient. In the group of acutely infected COVID-19 patients, all but one patient recognized at least two peptide specificities of the N protein with an average number of 12.44 (range 0–19; median 16) CD4+ T cell responses (**Fig 1B**).

One resolved patient (rCov-09) and one acute patient (aCov-07) did not show a M protein peptide-specific CD4+ T cell response. Of the other patients, the resolved patient group had an average of 6.96 (range 1–22; median 6) M-specific CD4+ T cell responses (**Fig 3**). The acute patient group showed an average of 6.2 (range 0–12, median 5.5) individual M peptide specific CD4+ T cell responses.

The pattern of the CD4+ T cell response against the overlapping peptide set of the smaller E protein is shown in **S3 Table**. From the resolved patient group, 13 patients (54.2%; n = 24) showed a detectable response against E, whereas four of the acutely infected patients showed a peptide specific CD4+ T cell response (40%; n = 10). An average of 1.67 responses per resolved

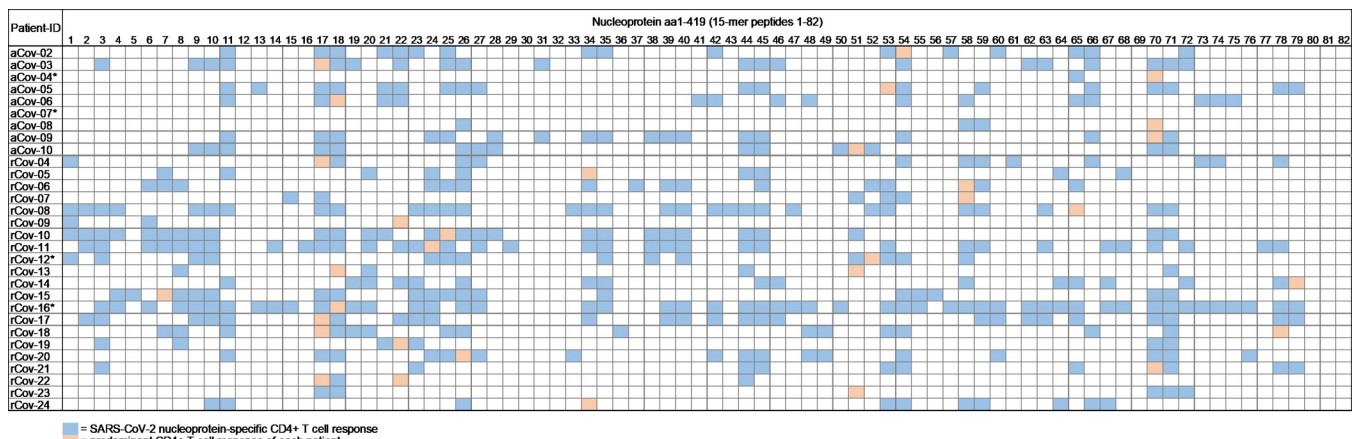

**Fig 2. Overview of the SARS-CoV-2 nucleoprotein-specific CD4+ T cell responses in patients with acute (n = 9) and resolved (n = 21) COVID-19.** For each patient, the HLA class II molecules DRB1 and DQB1 are listed. aCov = acute COVID-19 patient, rCov = resolved COVID-19 patient. * = patient was treated with immunosuppressant medication or received chemotherapy.

patient were detected (range 0–6) and in the acute patients, we saw an average of 0.8 CD4+ responses (range 0–3).

Overall, 84% of the 135 overlapping SARS-CoV-2 peptides (116/135) elicited at least one virus-specific CD4+ T cell response with 23.2 CD4+ T cell responses/ per resolved patient (total number of 557 CD4+ T cell responses) and 18.2 CD4+ T cell responses /per acute patient (total number of 182 CD4+ T cell responses) (**Fig 4**).

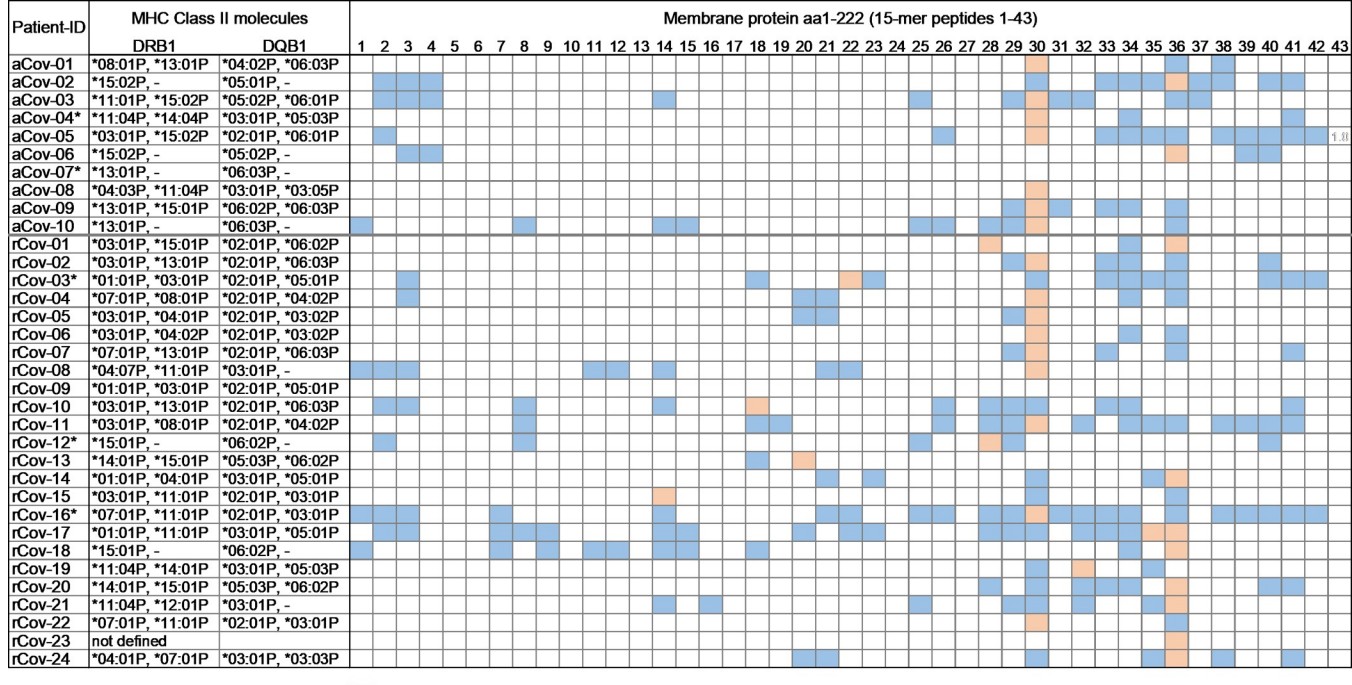

**Fig 3. Overview of the SARS-CoV-2 membrane protein-specific CD4+ T cell responses in patients with acute (n = 10) and resolved (n = 24) COVID-19.** For each patient, the HLA class II molecules DRB1 and DQB1 are listed. aCov = acute COVID-19, rCov = resolved COVID-19. * = patient was treated with immunosuppressant medication or received chemotherapy.

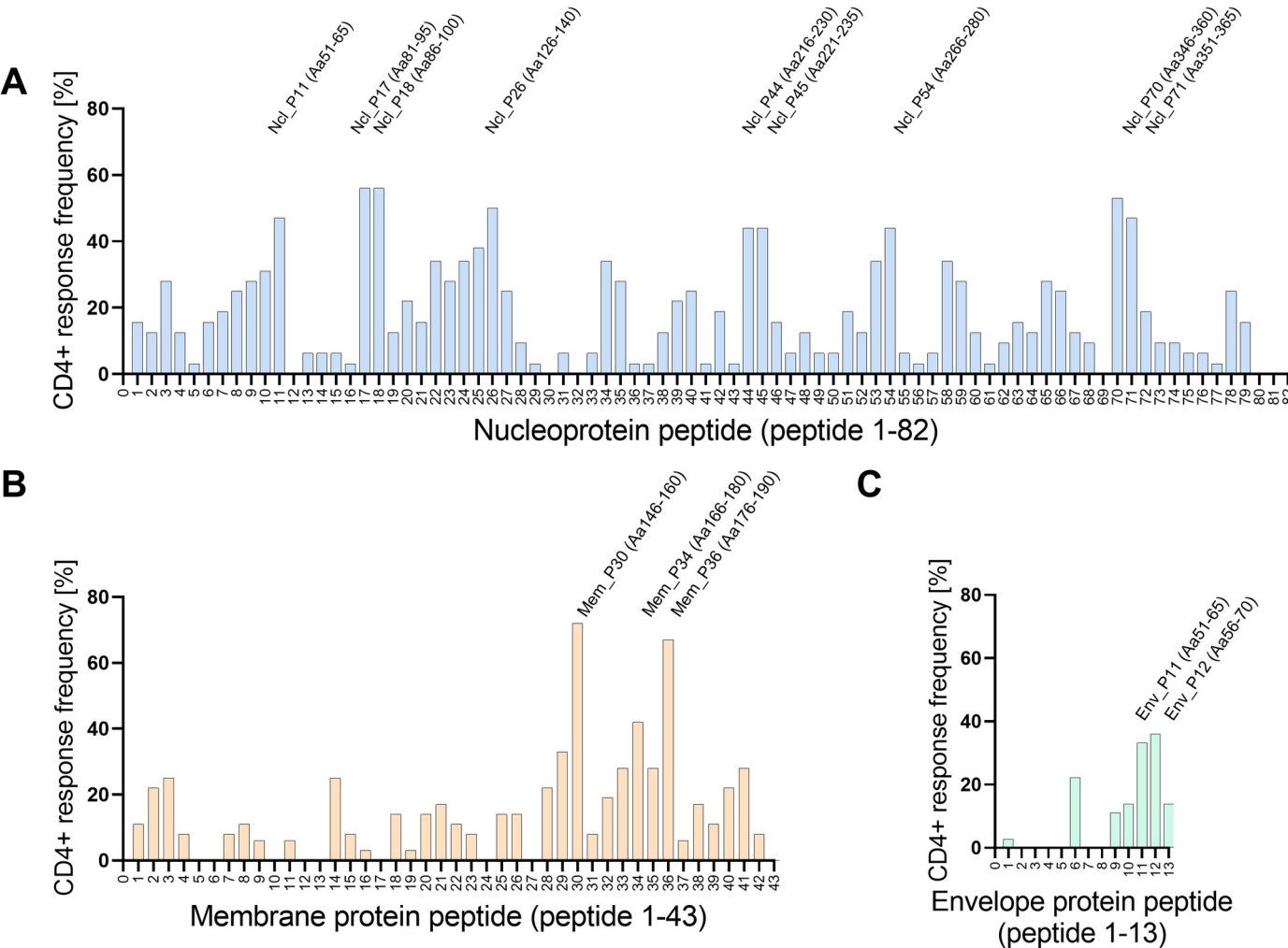

**Fig 4. Distribution of CD4+ T cell responses against the nucleo, membrane and envelope-protein. (A)** Overview of the nucleoprotein-specific CD4+ T cell response against 82 single nucleoprotein peptides. A total number of 30 patients were tested. **(B)** Overview of the SARS-CoV-2 membrane protein-specific CD4+ T cell response against 43 single membrane peptides. A total number of 34 patients were tested. **(C)** Overview of the SARS-CoV-2 envelope protein-specific CD4+ T cell response against 13 single envelope peptides. A total number of 34 patients were tested. Response frequency (RF) was calculated by dividing the number of patients that had a specific T cell response against a single peptide by the total number of patients who were tested.

In **S2 Fig**, all of the detected peptide-specific CD4+ T cell responses of patient rCov-17 are shown. This patient with resolved COVID-19 infection (129 days after first positive PCR) showed 26 CD4+ T cell responses against the N protein, 18 against the M protein and five against the E protein.

All but one patient responded to at least one peptide located within the N, M or E antigen (97%, 33/34). Patient aCov-07 who did not show a T cell response had an acute asymptomatic infection, was treated for a high-grade sarcoma with Doxorubicin/Ifosphamid, and was tested early into the infection (two days after the first positive PCR test).

In our healthy control group (n = 12), five subjects showed a N, M or E-specific CD4+ T cell response (42%) **(S4 Table)** with a mean of 1.4 CD4+ T cell responses per healthy control (range 0–9). Of note, on average these responses had a lower magnitude than peptide-specific T cell responses measured in the patient cohort.

The response frequency (RF), meaning the number of CD4+ T cell responses divided by the number of patients tested, was assessed for every peptide of the N, M and E protein (**Tables 2–4**). Tables **2–4** show all peptides with a RF above 20% in this study. The two most frequently detected peptides are Mem_P30 and Mem_P36, both located in the endodomain of the M protein [4]. 26 patients (RF: 72.2%; 26/34) showed a CD4+ T cell response against Mem_P30 (aa146-160), and 24 patients responded to Mem_P36 (aa176-190) (RF: 66.7%, 14/34). Within the N protein, Ncl_P17 (aa81-95) and Ncl_P18 (aa86-100) showed the highest RF of 56%. These two peptides are located within the N-terminal domain of the protein [24]. A number of epitopes described in the current study, have previously been reported (**Table 5**) [15, 25–28]. The RF from the respective previous reports as well as the HLA restriction are shown. From the peptides listed in **Tables 2–4**, eleven peptides have not been published before according to the IEDB database (www.iedb.org), including peptide Mem_P02, Ncl_P24, Ncl_P34, Ncl_P10, Ncl_P09, Ncl_P23, Ncl_P35, Ncl_P08, Ncl_P40, Ncl_P39, Env_P11.

While the *in vitro* cultivation of PBMC with 15-mer peptides is potentially biased towards the enrichment of antigen-specific CD4+ T cells, we also analyzed the peptide specific CD8+ T cell response (**S5 Table**). 70% of the acute patients (7/10) and 79% of the resolved patients (19/24) showed a N, M or E-specific CD8+ T cell response in this study. The highest number of CD8+ T cell responses were directed against peptide Mem_P30 with a total number of nine responses (RF: 26.5%). Most CD8+ T cell responses coincided with a parallel SARS-CoV-2-specific CD4+ T cell response and the average magnitude of these CD8+ T cell responses was smaller compared to the average peptide specific CD4+ T cell response.

## Correlation of the virus-specific CD4+ T cell response pattern with patient characteristics, clinical course and previous exposure to commonly circulating coronaviruses (CCC)

It should be noted that in contrast to previous studies, no significant correlation was found between age or gender and the N, M and E-specific CD4+ T cell response using our assays (**S3E Fig**). Similarly, no correlation was found between acute and convalescence status of the infection (**Fig 1B**) or the time after infection and the breadth or pattern of the virus-specific CD4+ T cell response (**S3A Fig**). The disease severity (mild/moderate against severe/critical)

**Table 2. Overview of the most frequently detected peptides of the SARS-CoV-2 nucleoprotein by CD4+ T cells in the current study.** Peptides detected in the current study with a response frequency (RF) ≥ 20%. Red and bold font indicates the optimal epitope length validated by truncation experiments for the peptides Mem_P30, Mem_P36 and Ncl_P18.

| Peptide | aa position | Sequence | | | | | | | | | | | | | | | RF | Peptide | aa position | Sequence | | | | | | | | | | | | | | | RF |
|---------|-------------|---|---|---|---|---|---|---|---|---|---|---|---|---|---|---|-----|---------|-------------|---|---|---|---|---|---|---|---|---|---|---|---|---|---|---|-----|
| Ncl_P17 | 81–95 | D | D | Q | I | G | Y | Y | R | R | A | T | R | R | I | R | 56% | Ncl_P58 | 286–300 | F | G | D | Q | E | L | I | R | Q | G | T | D | Y | K | H | 34% |
| Ncl_P18 | 86–100 | **Y** | **Y** | **R** | **R** | **A** | **T** | **R** | **R** | **I** | **R** | G | G | D | G | K | 56% | Ncl_P10 | 46–60 | P | N | N | T | A | S | W | F | T | A | L | T | Q | H | G | 31% |
| Ncl_P70 | 346–360 | F | K | D | Q | V | I | L | L | N | K | H | I | D | A | Y | 53% | Ncl_P03 | 11–25 | N | A | P | R | I | T | F | G | G | P | S | D | S | T | G | 28% |
| Ncl_P26 | 126–140 | N | K | D | G | I | I | W | V | A | T | E | G | A | L | N | 50% | Ncl_P09 | 41–55 | R | P | Q | G | L | P | N | N | T | A | S | W | F | T | A | 28% |
| Ncl_P11 | 51–65 | S | W | F | T | A | L | T | Q | H | G | K | E | D | L | K | 47% | Ncl_P23 | 111–125 | Y | Y | L | G | T | G | P | E | A | G | L | P | Y | G | A | 28% |
| Ncl_P71 | 351–365 | I | L | L | N | K | H | I | D | A | Y | K | T | F | P | P | 47% | Ncl_P35 | 171–185 | F | Y | A | E | G | S | R | G | G | S | Q | A | S | S | R | 28% |
| Ncl_P44 | 216–230 | D | A | A | L | A | L | L | L | L | D | R | L | N | Q | L | 44% | Ncl_P59 | 291–305 | L | I | R | Q | G | T | D | Y | K | H | W | P | Q | I | A | 28% |
| Ncl_P45 | 221–235 | L | L | L | L | D | R | L | N | Q | L | E | S | K | M | S | 44% | Ncl_P65 | 321–335 | G | M | E | V | T | P | S | G | T | W | L | T | Y | T | G | 28% |
| Ncl_P54 | 266–280 | K | A | Y | N | V | T | Q | A | F | G | R | R | G | P | E | 44% | Ncl_P08 | 36–50 | R | S | K | Q | R | R | P | Q | G | L | P | N | N | T | A | 25% |
| Ncl_P25 | 121–135 | L | P | Y | G | A | N | K | D | G | I | I | W | V | A | T | 38% | Ncl_P27 | 131–145 | I | W | V | A | T | E | G | A | L | N | T | P | K | D | H | 25% |
| Ncl_P22 | 106–120 | P | R | W | Y | F | Y | Y | L | G | T | G | P | E | A | G | 34% | Ncl_P40 | 196–210 | N | S | T | P | G | S | S | K | R | T | S | P | A | R | M | 25% |
| Ncl_P24 | 116–130 | G | P | E | A | G | L | P | Y | G | A | N | K | D | G | I | 34% | Ncl_P66 | 326–340 | P | S | G | T | W | L | T | Y | T | G | A | I | K | L | D | 25% |
| Ncl_P34 | 166–180 | T | L | P | K | G | F | Y | A | E | G | S | R | G | G | S | 34% | Ncl_P78 | 386–400 | Q | K | K | Q | Q | T | V | T | L | L | P | A | A | D | L | 25% |
| Ncl_P53 | 261–275 | K | R | T | A | T | K | A | Y | N | V | T | Q | A | F | G | 34% | Ncl_P20 | 96–110 | G | G | D | G | K | M | K | D | L | S | P | R | W | Y | F | 22% |

**Table 3. Overview of the most frequently detected peptides of the SARS-CoV-2 membrane protein by CD4+ T cells in the current study.** Peptides detected in the current study with a response frequency (RF) $\geq$ 20%. Red and bold font indicates the optimal epitope length validated by truncation experiments for the peptides Mem_P30, Mem_P36 and Ncl_P18.

| Peptide | aa position | Sequence | | | | | | | | | | | | | | RF |
|---|---|---|---|---|---|---|---|---|---|---|---|---|---|---|---|---|
| Mem_P30 | 146–160 | R | G | **H** | **L** | **R** | **I** | **A** | **G** | **H** | **H** | **L** | **G** | **R** | C | D | 72% |
| Mem_P36 | 176–190 | L | S | **Y** | **Y** | **K** | **L** | **G** | **A** | **S** | **Q** | **R** | **V** | **A** | G | D | 67% |
| Mem_P34 | 166–180 | K | E | I | T | V | A | T | S | R | T | L | S | Y | Y | K | 42% |
| Mem_P29 | 141–155 | G | A | V | I | L | R | G | H | L | R | I | A | G | H | H | 33% |
| Mem_P33 | 161–175 | I | K | D | L | P | K | E | I | T | V | A | T | S | R | T | 28% |
| Mem_P35 | 171–185 | A | T | S | R | T | L | S | Y | Y | K | L | G | A | S | Q | 28% |
| Mem_P41 | 201–215 | I | G | N | Y | K | L | N | T | D | H | S | S | S | S | D | 28% |
| Mem_P03 | 11–25 | E | E | L | K | K | L | L | E | Q | W | N | L | V | I | G | 25% |
| Mem_P14 | 66–80 | V | L | A | A | V | Y | R | I | N | W | I | T | G | G | I | 25% |
| Mem_P02 | 6–20 | G | T | I | T | V | E | E | L | K | K | L | L | E | Q | W | 22% |
| Mem_P28 | 136–150 | S | E | L | V | I | G | A | V | I | L | R | G | H | L | R | 22% |
| Mem_P40 | 196–210 | Y | S | R | Y | R | I | G | N | Y | K | L | N | T | D | H | 22% |

in this small cohort also did not correlate with the breadth of the CD4+ T cell response in this study (**S3B Fig**). With the exception of patients aCov-07 (see above), the four other immuno-suppressed patients including patient rCov-12 and rCov-16, who were both B cell-depleted, showed a comparable number of T cell responses (**Fig 1B**) [30].

The aim of this study was to define dominant SARS-CoV-2-specific CD4+ T cell epitopes located within the N, M and E protein. Larger studies are necessary to investigate correlations between the clinical status and the SARS-CoV-2-specific T cell response.

To investigate the possibility of cross-reactivity with CCC, we analyzed sequence homology of the most frequently detected SARS-CoV-2 epitopes in this study with the CCC 229E, HKU1, OC43, and NL63 (**S6 Table**) [27]. Overall, sequence homology was low to intermediate (range 27–87%) with the highest sequence homology located in Ncl_P22 (range: 60–87%). Interestingly, this peptide was detected by two of our healthy controls (HC-06, HC-12).

To investigate past exposure to CCC, we measured IgG antibody responses against 229E, HKU1, OC43, and NL63 and SARS-CoV-2 in the frozen plasma from 30 patients in our cohort as well as four healthy controls using a commercial recomLine SARS-CoV-2 IgG Immunoblot assay (**S7 Table**). All healthy controls were seronegative for SARS-CoV-2 antibodies. In our patient cohort five out of ten (50%) acutely infected patients had detectable IgG antibodies, whereas in the resolved patient group all but one patient showed antibody responses (95%). The one patient who did not show an antibody response was patient rCov-12 who was B-cell depleted due to Rituximab treatment. The prevalence of antibodies to CCC or SARS-CoV-2 did not influence the breadth of antigen-specific CD4+ T cell responses detected in this study (**S3C and S3D Fig**).

**Table 4. Overview of the most frequently detected peptides of the SARS-CoV-2 envelope protein by CD4+ T cells in the current study.** Peptides detected in the current study with a response frequency (RF) $\geq$ 20%.

| Peptide | aa position | Sequence | | | | | | | | | | | | | | RF |
|---|---|---|---|---|---|---|---|---|---|---|---|---|---|---|---|---|
| Env_P12 | 56–70 | F | Y | V | Y | S | R | V | K | N | L | N | S | S | R | V | 36% |
| Env_P11 | 51–65 | L | V | K | P | S | F | Y | V | Y | S | R | V | K | N | L | 33% |
| Env_P06 | 26–40 | F | L | L | V | T | L | A | I | L | T | A | L | R | L | C | 22% |

## Fine mapping and restriction experiments

The optimal length of the epitopes Mem_P30 (aa146-160), Mem_P36 (aa176-190) and Ncl_P18 (aa86-100) was assessed by additional experiments using peptide truncations. PBMC of patients aCov-03 and rCov-17, who had shown a strong CD4+ T cell response against Mem_P30, Mem_P36 and Ncl_P18 were cultivated with the respective peptides and their corresponding set of truncations. For the truncations, peptides were synthesized either shortened at the N- or the C-terminus by two or four amino acids, respectively (**Fig 5A**). Both patients showed a similar distribution of IFN-γ responses across all truncations and both patients expressed HLA-DRB1*11:01, which is likely one of the restricting HLA molecule for all three peptides. This was supported by the *in vitro* data (**Tables 6 and 7**) that showed binding to HLA-DRB1*11:01 with high affinity of all three peptides Mem_P30, Mem_P36 and Ncl_P18. Interestingly, healthy control HC-06 responded to Ncl_P17 and Ncl_P18 and also expressed HLA-DRB1*11:01. To confirm restriction of Mem_P30, Mem_P36 and Ncl_P18, we used PBMC as antigen presenting cells (APC) from healthy donors who expressed the respective HLA-molecule (**Fig 5B**). Original FACS-plots are shown in **S4 Fig**. Mem_P30, and Ncl_P18 were most pronouncedly presented by matched and peptide-pulsed PBMC expressing HLA-DRB1*11:01, and Mem_P36 by PBMC expressing DRB1*01:01.

The *in vitro* HLA class II binding data (**Tables 6 and 7**) indicated that the peptides Env_P11, Mem_P30, Mem_P34, Mem_P36 and Ncl_P70 peptides that were most frequently recognized in this study (RF range: 36.1–72.2%) had the capacity to bind 12 or more of the 22 HLA-molecules tested with an affinity of 1000 nM or better.

We then compared the pattern of *in vitro* HLA binding with the HLA-molecules expressed by the patients from this study. The most likely binding HLA-DRB1 molecules are shown in **S8 Table**. The Mem_P30 peptide was consistently recognized by all five patients expressing HLA molecule DRB1*07:01 and all six expressing DRB1*11:01 (**S8 Table**). Correspondingly, the *in vitro* binding data revealed that the peptide bound both molecules with high affinity (132 nM and 34 nM, respectively). Furthermore, Mem_P30 is also likely restricted by DRB1*04:01, given that it binds the molecule with an affinity of 998nM, and all three DRB1*04:01 positive patients responded to this peptide.

The Mem_P36 peptide was found to bind DRB1*01:01 with very high affinity (0.83 nM) and elicited a CD4+ T cell response in three out of four DRB1*01:01 patients. Similarly, Mem_P36 is likely restricted by DRB1*04:01, 07:01, 11:01 and 15:01 (**S8 Table**).

The Ncl_17 and Ncl_18 peptides, which where both recognized by all six patients with DRB1*11:01, bound DRB1*11:01 with very high affinities of 20 and 71 nM, respectively (**Tables 7 and S8**). These *in vitro* data are in concordance with the restriction data that was previously published (**Table 5**).

A number of directly neighboring peptides elicited a similar CD4+ T cell response in individual patients, for example in Ncl_P17 and P18 (**Fig 4**). Truncation 3 of Ncl_P18 is shortened towards Ncl_P17 and did not elicit a CD4+ T cell response (**Fig 5A**). This indicates that the epitope is most likely located between the overlapping peptides Ncl_P17 and P18.

## *Ex vivo* CD4+ T cell phenotype

We stimulated PBMC from six acute COVID-19 patients with the ten most frequently detected peptides (listed in **Table 5**), in order to investigate the frequency and functionality of the *ex vivo* SARS-CoV-2-specific CD4+ T cell response. The flow cytometry panel is shown in **S9 Table**. The *ex vivo* cytokine response was measured by ICS and the results are shown in **S5A and S5B Fig**. Due to their relatively low frequency, SARS-CoV-2-specific CD4+ T cells were difficult to assess *ex vivo*. However, three of six acute patients showed an IFN-γ response above

**Table 5. Overview of the most frequently detected peptides of the SARS-CoV-2 nucleo, membrane and envelope-protein by CD4+ T cells in previously published studies.**

| Protein | aa position | Sequence | RF in previous study | Peptide in this study | HLA restriction | Reference |
|---|---|---|---|---|---|---|
| Membrane protein | 145–160 | R G H L R I A G H H L G R C D | 46% | Mem_P30 | DRB1*15:01, DRB1*14:01, DRB1*13:01, DRB1* 07:01, DRB1*01:01, DRB1*07:01 | [25, 29] |
| | 175–190 | L S Y Y K L L G A S Q R V A G D | 56% | Mem_P36 | DQB1*03:01, DRB1*16:02, DRB1*16:01, DQB1*06:03, DQB1*06:02, DRB1*15:01, DRB1*14:06 | [25, 26, 28] |
| Nucleo-protein | 50–65 | S W F T A L T Q H G K E D L K | 20% | Ncl_P11 | not defined | [25, 29] |
| | 80–95 | D D Q I G Y Y R R A T R R I R | 6% | Ncl_P17 | DRB1*13:01, DRB1*15:01, DRB1*14:06, DRB1*14:01, DRB1*16:02 | [25, 27, 29] |
| | 85–100 | Y Y R R A T R R I R G G D G K | 23% | Ncl_P18 | DRB1*13:01, DRB1*14:06 | [25] |
| | 125–140 | N K D G I I W V A T E G A L N | 29% | Ncl_P26 | DQB1*02:01, DRB1*07:01, DRB1*04:04, DQB1*05:03, DQB1*02:02, DQB1*03:02 | [25] |
| | 215–230 | D A A L A L L L D R L N Q L | 14% | Ncl_P44 | DQB1*05:03, DRB1*14:01, DRB1*03:01, DRB1*05:03, DRB1*15:01, DRB1*12:01 | [25, 27] |
| | 345–360 | F K D Q V I L N K H I D A Y | 15% | Ncl_P70 | DRB1*16:02, DRB1*15:01, DQB1*06:02, DRB1*14:06, DRB1*03:01 | [25] |
| | 350–365 | I L L N K I D A Y K T K T F P | 24% | Ncl_P71 | DRB1*14:06, DRB1*15:01, DRB1*14:01 | [25] |
| Envelope protein | 55–70 | F Y V Y S R V K N L N S S R V | 31% | Env_P12 | not defined | [26] |

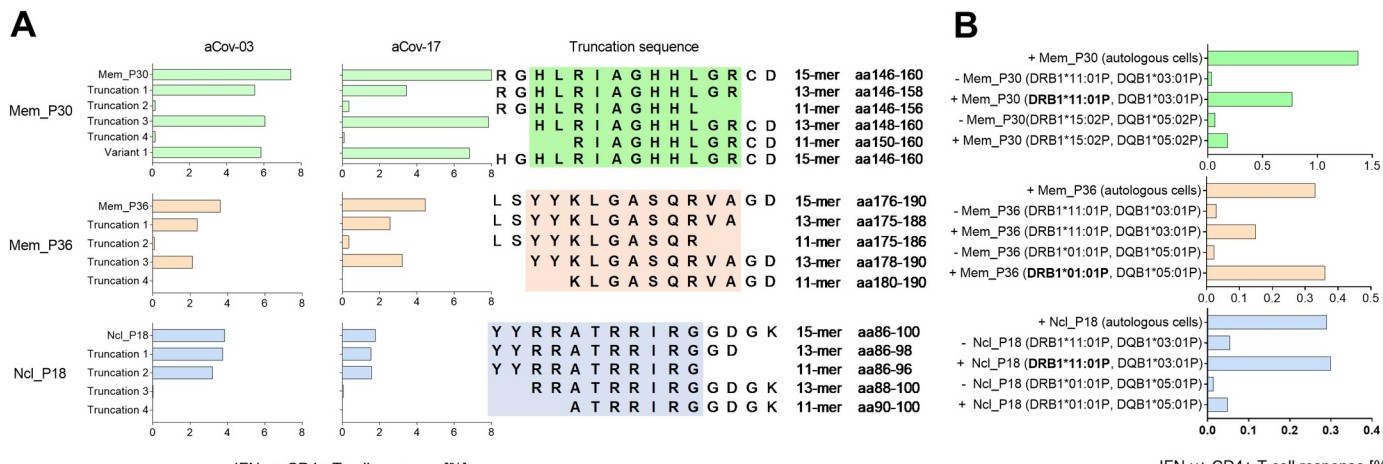

**Fig 5. Fine mapping of the most frequently detected peptides Mem_P30 (aa146-160), Mem_P36 (aa176-190) and Ncl_P18 (aa86-100).** (A) Truncation experiments of Mem_P30, Mem_P36 and Ncl_P18 in patients aCov-03 and rCov-17. For Mem_P30 one variant that contained the amino acid H instead of R was also tested. (B) Restriction experiments of Mem_P30, Mem_P36 and Ncl_P18 in patients aCov-03 and rCov-17. PBMC from matched donors were used as antigen-presenting cells. Autologous cells served as a positive control. Most likely HLA-restriction is indicated by bold font.

the detection level (detection level: 0.02% of all CD4+ T cells for IFN-y, Il-2, and TNF-a). The responses ranged from 0% - 0.1% (IFN-y), 0.01% - 0.21% (TNF-α) and 0% - 0.08% (IL-2) of CD4+ T cells. CD4+ T cells predominantly produced TNF-α and the frequencies of multifunctional IFN-y+TNFα+IL-2+ CD4+ T cells was low. No Th2 cytokines such as IL-4 were assessed. PBMC of two resolved COVID-19 patients were also stimulated with the SARS-CoV-2 peptide pool but no *ex vivo* CD4+ T cell response could be detected.

Based on our experimental epitope truncation and restriction data (Fig 5), we were able to develop an M-specific DRB*11-restricted MHC class II tetramer containing the 20-mer peptide sequence Mem_aa145-164 (LRGHLRIAGHHLGRCDIKDL) containing Mem_P30, the most frequently detected peptide in this study. We performed an *ex vivo* staining of PBMC

**Table 6. *In vitro* binding capacity of 14 SARS-CoV-2-specific peptides to 22 frequent HLA class II MHC molecules.** Binding capacities are expressed as IC50 nM values measured in classical *in vitro* binding assays based on inhibition of binding of a high affinity radiolabeled ligand to purified HLA molecules. High affinity binding is defined as IC50 < 1,000 nM and highlighted by bold font. For reasons of comprehensibility, values larger than 40,000 nM are indicated by a dash. The total number of alleles bound as well as the response frequency (RF) of the responding peptide is shown. Mem_P30 and Mem_P36, with the highest RFs, bound to 12 and 13 HLA molecules, respectively.

| Peptide | aa position | | | | | | | | | | | | | | | Alleles bound | RF | DPB1*02:01 | DPA1*01:03 DPB1*04:01 | DQA1*05:01 DQB1*02:01 | DQA1*05:01 DQB1*03:0 | DQA1*03:01 DQB1*03:02 | DQA1*01:01 DQB1*05:01 | DQA1*01:02 DQB1*06:02 |
|---|---|---|---|---|---|---|---|---|---|---|---|---|---|---|---|---|---|---|---|---|---|---|---|---|
| Env_11 | 46–60 | L | V | K | P | S | F | Y | V | Y | S | R | V | K | N | L | 13 | 36.1 | **42** | **63** | - | 5766 | - | - | 24387 |
| Mem_30 | 146–160 | R | G | H | L | R | I | A | G | H | H | L | G | R | C | D | 12 | 72.2 | **569** | **445** | - | 15141 | - | 37659 | - |
| Mem_34 | 166–180 | K | E | I | T | V | A | T | S | R | T | L | S | Y | Y | K | 13 | 41.7 | 1909 | 3337 | 13645 | **649** | 17391 | - | 1827 |
| Mem_36 | 176–190 | L | S | Y | Y | K | L | G | A | S | Q | R | V | A | G | D | 13 | 66.7 | 13822 | - | 22448 | **48** | - | 15193 | **428** |
| Ncl_11 | 51–65 | S | W | F | T | A | L | T | Q | H | G | K | E | D | L | K | 9 | 46.9 | - | 35760 | 30440 | 1304 | 19748 | - | **952** |
| Ncl_17 | 81–95 | D | D | Q | I | G | Y | Y | R | R | A | T | R | R | I | R | 7 | 56.3 | 34655 | - | - | 5067 | - | - | - |
| Ncl_18 | 86–100 | Y | Y | R | R | A | T | R | R | I | R | G | G | D | G | K | 6 | 56.3 | - | - | - | 6720 | - | - | - |
| Ncl_22 | 106–120 | P | R | W | Y | F | Y | Y | L | G | T | G | P | E | A | G | 5 | 34.4 | 10053 | - | **79** | 1278 | **414** | 2239 | 7882 |
| Ncl_25 | 121–135 | L | P | Y | G | A | N | K | D | G | I | I | W | V | A | T | 2 | 37.5 | 13013 | 20045 | 11706 | 14585 | 3795 | - | **203** |
| Ncl_26 | 126–140 | N | K | D | G | I | I | W | V | A | T | E | G | A | L | N | 9 | 50 | 4140 | 8348 | **422** | 4027 | **103** | - | 2643 |
| Ncl_44 | 216–230 | D | A | A | L | A | L | L | L | D | R | L | N | Q | L | | 7 | 43.8 | 1162 | 2779 | 2487 | 22499 | **860** | 8458 | 2499 |
| Ncl_45 | 221–235 | L | L | L | L | D | R | L | N | Q | L | E | S | K | M | S | 9 | 43.8 | **285** | **701** | 3315 | - | 11904 | 23089 | 8081 |
| Ncl_54 | 266–280 | K | A | Y | N | V | T | Q | A | F | G | R | R | G | P | E | 9 | 43.8 | 8438 | 6056 | - | **794** | 33174 | - | 2936 |
| Ncl_70 | 346–360 | F | K | D | Q | V | I | L | L | N | K | H | I | D | A | Y | 13 | 53.1 | 287 | **794** | 30656 | 33658 | 9176 | 6010 | 8602 |

**Table 7. _In vitro_ binding capacity of 14 SARS-CoV-2-specific peptides to 22 frequent HLA class II MHC molecules.** Binding capacities are expressed as IC50 nM values measured in classical _in vitro_ binding assays based on inhibition of binding of a high affinity radiolabeled ligand to purified HLA molecules. High affinity binding is defined as IC50 < 1,000 nM and highlighted by bold font. For reasons of comprehensibility, values larger than 40,000 nM are indicated by a dash. The total number of alleles bound as well as the response frequency (RF) of the responding peptide is shown. Mem_P30 and Mem_P36, with the highest RFs, bound to 12 and 13 HLA molecules, respectively.

| Peptide | aa position | Sequence | Alleles bound | RF | DRB1*01:01 | DRB1*03:01 | DRB1*04:01 | DRB1*04:05 | DRB1*07:01 | DRB1*08:02 | DRB1*09:01 | DRB1*11:01 | DRB1*12:01 | DRB1*13:02 | DRB1*15:01 | DRB3*01:01 | DRB3*02:02 | DRB4*01:01 | DRB5*01:01 |
|---|---|---|---|---|---|---|---|---|---|---|---|---|---|---|---|---|---|---|---|
| Env_11 | 46–60 | L V K P S F Y V V Y S R V K N L | 13 | 36.1 | **13** | - | 869 | 470 | **5.3** | **27** | **52** | **14** | 1340 | 7253 | **78** | **104** | **160** | 38933 | **7.5** |
| Mem_30 | 146–160 | R G H L R I A G H H L G R R C D | 12 | 72.2 | **59** | 14579 | **998** | 7618 | **132** | **65** | **92** | **34** | 1232 | **900** | **55** | 1277 | 1458 | **159** | **24** |
| Mem_34 | 166–180 | K E I T V A T S R T L S Y Y Y K | 13 | 41.7 | **9.3** | **416** | **341** | 3904 | **38** | **228** | **17** | **14** | **174** | **205** | **6.2** | 5959 | 18244 | **770** | **41** |
| Mem_36 | 176–190 | L S Y Y K L G A S Q R V A G D D | 13 | 66.7 | **0.83** | 28371 | **21** | **63** | **5.3** | **74** | **9** | **187** | 16415 | 2734 | **91** | **169** | **28** | 26279 | **1.2** |
| Ncl_11 | 51–65 | S W F T A L T Q H G K E D L K | 9 | 46.9 | **40** | - | **192** | 875 | 715 | 795 | 445 | 493 | 7871 | - | 4938 | - | 5940 | 13917 | **13** |
| Ncl_17 | 81–95 | D D Q I G Y T Q R R A T R I R | 7 | 56.3 | **43** | - | 8423 | 9123 | **111** | **17** | 565 | **20** | - | - | 340 | 1029 | 1023 | - | **2.7** |
| Ncl_18 | 86–100 | Y Y R R A T R I R G G D G K | 6 | 56.3 | **62** | - | 9044 | 13721 | **111** | **78** | 732 | **71** | 16525 | - | 1388 | 5089 | 8484 | 9570 | **16** |
| Ncl_22 | 106–120 | P R W Y F Y Y L G T G P E A G | 5 | 34.4 | **8.5** | - | 3013 | **18** | 3633 | 1226 | 1131 | 3391 | 23302 | - | 775 | 2770 | 3172 | - | 1097 |
| Ncl_25 | 121–135 | L P Y G A N K D G I I W V A T | 2 | 37.5 | 1310 | 39342 | 20706 | 7532 | 1475 | 10551 | 3830 | 13942 | 19091 | - | **937** | 13782 | 6365 | 10562 | 1667 |
| Ncl_26 | 126–140 | N K D G I I W V A T E G A L N | 9 | 50 | **42** | 4705 | **184** | 3688 | **219** | 6498 | **141** | 9109 | - | - | **596** | 14419 | 2483 | **843** | 735 |
| Ncl_44 | 216–230 | D A A L A L L L D R L L N Q L | 7 | 43.8 | 9037 | 2446 | 6091 | **840** | - | 1141 | 12530 | **317** | **824** | **486** | 7139 | **85** | - | **43** | 26925 |
| Ncl_45 | 221–235 | L L L L D R L N Q L E S K M S | 9 | 43.8 | **95** | 1195 | 9776 | 1852 | 12332 | **591** | 8386 | **94** | **481** | **673** | **134** | **135** | - | 1702 | 6347 |
| Ncl_54 | 266–280 | K A Y N V T Q A F G R R G P E | 9 | 43.8 | **12** | - | **647** | 11692 | **29** | **389** | 105 | **196** | - | 6728 | **988** | 7837 | 3663 | 2488 | **0.75** |
| Ncl_70 | 346–360 | F K D Q V I L L N K H I D A Y | 13 | 53.1 | **42** | 6458 | 14275 | 1318 | 620 | **152** | 1559 | **96** | **89** | **187** | **68** | **309** | **205** | **719** | **168** |

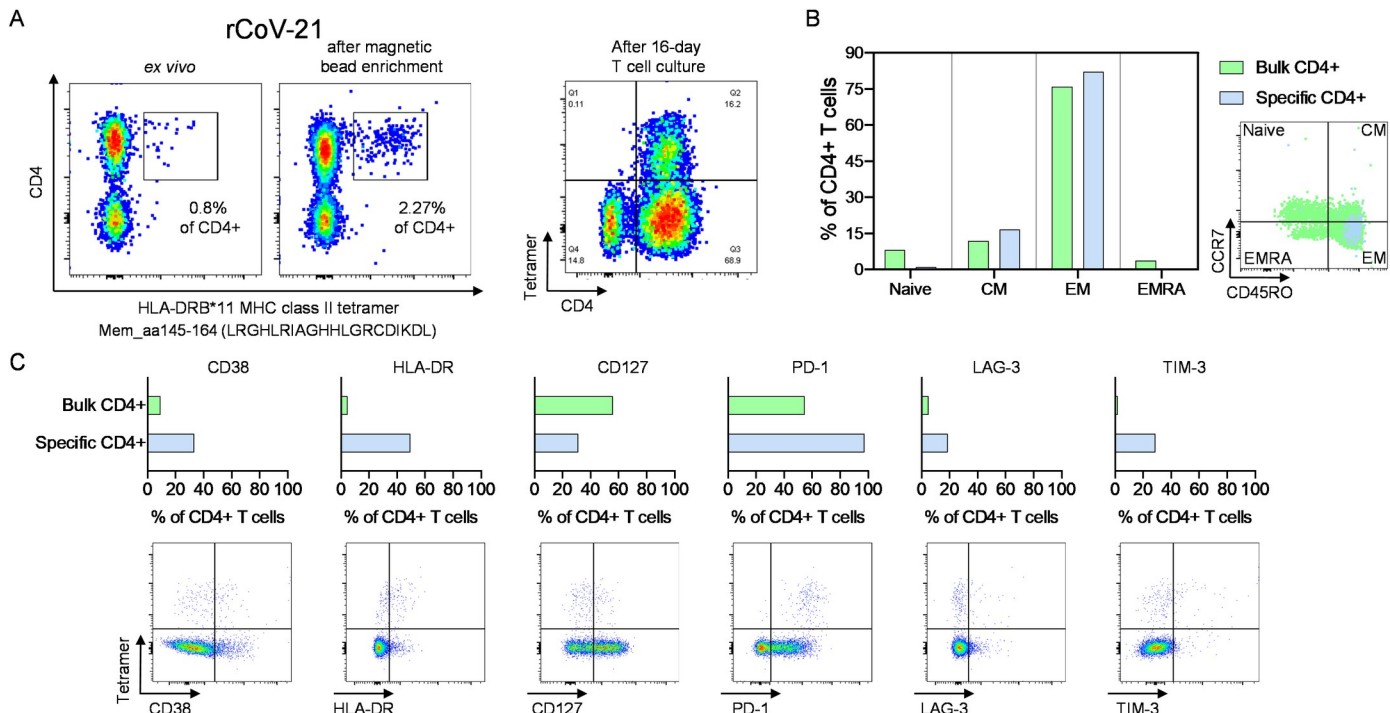

**Fig 6. The *ex vivo* peptide-specific CD4+ T cell response in patient rCov21. (A)** Frequency of tetramer+CD4+ T cells *ex vivo*, after magnetic bead enrichment, and after 16-day expansion with the corresponding peptide and IL-2. **(B)** Tetramer+CD4+ T cells differentiated to an effector and central memory type. **(C)** Antigen-specific CD4+ T cells show an activated and exhausted phenotype. CM = central memory, EM = effector memory, EMRA = terminally differentiated effector memory cells.

from patient rCov21 (DRB1*07:01P, *11:01P) who had the corresponding HLA-type and who had shown a CD4+ T cell response against Mem_P30 (**Fig 6A**). The patient had a severe course of COVID-19, and the blood sample used for this MHC class II tetramer staining was collected 20 days after the first positive PCR. The antigen-specific tetramer+ CD4+ T cells were mainly effector memory cells (EM) and showed upregulation of the activation molecules CD38 and HLA-DR and low CD127 expression, as well as ubiquitous expression of PD-1 and slight upregulation of LAG-3 and TIM-3 compared to bulk CD4+ cells (**Fig 6B and 6C**).

## Discussion

The main result of our study was the detection of a universally broad CD4+ T cell response specific for the N, M and to a lesser degree E-protein, regardless of the clinical course. With our sensitive *in vitro* testing approach, we were able to detect T cell responses in 33 of 34 patients (97%) and found 44 N, M and E-specific peptides with a RF of 20% or higher. One must keep in mind that the three peptide sets only covered a small number of the structural and non-structural proteins encoded by the SARS-CoV-2 genome [31].

To date, most studies have utilized pools of predicted peptides or pools of overlapping peptides spanning entire antigens, to probe responses to different SARS-CoV-2 antigens [3, 7–9, 12–14]. However, the exact T cell epitopes and immunodominant antigen regions have only been determined in few studies [15, 25–27]. Notably, these studies have overall reported a similar response pattern (**Table 5**)**,** suggesting a reproducibility despite differences in the patient cohorts and methodologies. The peptides described in **Tables 2–4** elicited a CD4+ T cell response in 22–72% of our study group that had an HLA-background representative of the

European population. On the other hand, several of the most prevalent alleles in the general worldwide population including DRB1*03:01, DRB1*04:01, DRB1*07:01, DRB1*11:01 and DRB1*15:01 [32], were also well represented in our cohort. However, similar studies should be conducted in varying populations.

The broad SARS-CoV-2-specific T cell response detected here, is seemingly primed at an early timepoint of infection, since we detected no significant difference between the number and magnitude of T cell responses in early acute vs. resolved infection (**Fig 1B**).

In further longitudinal studies, it will be important to expand our understanding of the *ex vivo* kinetics of T cell priming, the role of antigen-specific T cells for viral control, and the potential role of the antiviral T cell response in the pathogenesis of the cytokine storm [33–35]. Furthermore, the tissue-resident virus-specific T cell response will be of high interest [36]. MHC class I and II multimer technology using epitopes located within the different SARS-CoV-2 structural proteins will be another tool for these investigations [37].

We did not find any correlation between the virus-specific T cell response directed against the different structural proteins and clinical parameters such as gender, age, or disease severity (**S3 Fig**). Of note, our study was rather designed for epitope characterization than to detect differences between immunological and clinical status. At the same rate, the T cell pattern of this acute resolving viral infection is not as dichotomous as in a chronically evolving infection like HCV and the universally broad T cell response seems to be a characteristic for a SARS-CoV-2 infection [20].

We detected few CD4+ T cell responses with low magnitude in five out of twelve healthy controls. These responses that were measured after stimulation with SARS-Cov-2-specific peptide pools might possibly stem from cross-reactive memory responses originally primed against one or more of the CCC [37, 38]. The serological assessment of preexisting antibodies against CCC revealed that the majority of the samples tested, regardless of the infection status, had serologically been exposed to CCC (**S7 Table**). This did not seem to have influenced the breadth of the antigen-specific CD4+ T cell response as we did not find differences in the prevalence of preexisting antibodies and the number or distribution of SARS-CoV-2-specific CD4+ T cell responses using our methodology. Further studies with higher numbers of unexposed healthy controls, testing individual SARS-CoV-2-specific peptides are necessary to define cross-reactive immune epitopes [39].

Multiple SARS-CoV-2 variants are circulating globally [40], and the three variants Alpha (first emerged in the UK), Beta (South Africa), Gamma (Brazil), Delta (India) were labeled by the CDC as variants of concern (https://www.cdc.gov/coronavirus/2019-ncov/more/science-and-research/scientific-brief-emerging-variants.html). Most mutations of these three variants are located in the spike protein: The Gamma lineage for example contains three mutations in the spike protein receptor binding domain: K417T, E484K, and N501Y. Furthermore, there is evidence that the E484K mutation may affect neutralizing antibodies [41]. Interestingly, only few mutations are located in the N, M or E antigen (**S6 Fig**). The Beta lineage for example only shows one mutation in the E protein (P71L), and one in N (T205I), and Alpha and Gamma variant only show one mutation in the N protein, respectively (S235F, P80R), whereas the Delta variant contains three mutations in the N protein (D63G, R203M, D377Y) (https://cov-lineages.org/). With the extensive breadth of the CD4+ T cell response in each individual patient and low mutation rates of the N, M and E antigen, viral escape from the N, M or E-specific T cell response does not appear very likely [37]. However, the relative role of neutralizing antibodies and T cell responses for protection against symptomatic reinfection has not been defined for SARS-CoV-2. Of note, immunodominant regions priming virus-specific CD4+ T cells were reported to have minimal overlap with antibody epitopes [25].

Previously, a direct role of CD4+ T cells for anti-viral immunity has been suggested in animal models [42, 43] and in humans [7, 11, 13]. With the generated data, we were able to establish a M-specific MHC II tetramer which allowed us to perform an *ex vivo* analysis from one patient 20 days after diagnosis. We detected an *ex vivo* frequency of 0.8% of LRGHLRIAGHHLGRCDIKDL-specific CD4+ T cells that showed upregulation of several activation, and co-inhibitory molecules (**Fig 6**). With this MHC class II tetramer and further SARS-CoV-2-specific MHC II tetramers that will be established in due time, it will be easier to perform longitudinal and phenotypical analysis of antigen-specific CD4+ T cells during SARS-CoV-2 infection and will allow live-sort of antigen-specific CD4+ T cells for further functional assays.

Using a peptide pool with the most commonly detected peptides from this study, we were additionally able to detect virus-specific *ex vivo* T cell responses by IFN-y, TNF-α, and IL-2 cytokine staining (**S5 Fig**). Intriguingly, it has been reported that T cells with different SARS-CoV-2 epitope specificities have different phenotypes and functionalities: whereas S-specific CD4+ T cells were skewed towards a circulating T follicular helper profile, N and M-specific CD4+ T cells showed a Th1 or a Th1/Th17 profile [13].

It has already been reported that overlapping peptide pools of the M, N and E protein have been shown to induce SARS-CoV-2-reactive T cell responses with a relative dominance of CD4+ over CD8+ T cells [9, 13]. Most virus-specific CD8+ T cell responses detected in this study were weak and coincided with a CD4+ T cell response directed against the same peptide in the same patient. Accordingly, we did not follow-up on fine-mapping these subdominant CD8+ T cell responses.

Notably, the patients rCov-12 and rCov-16 were B cell-depleted but still exhibited a broad range of T cell responses—the results of the T cell assays of one B cell depleted patient were previously reported [30]. Despite a broad T cell response, patient rCov-16 showed a chronical course of SARS-CoV-2 infection while other patients were able to clear the virus [44]. It will be important to study the complex and complimentary roles of coordinated B and T cell responses in establishing viral clearance [7, 9]. We also know from previous studies that antibody responses against coronaviruses can be short-lived [45, 46]. Coronavirus-induced cellular immunity is predicted to be more sustained, and it would be highly interesting to further investigate in how far the SARS-CoV-2 specific T cell response alone can confer immunity. Additionally, it will be important to understand if a broad natural T cell memory response will render better protection than the vaccine-induced T cell response since most vaccines e.g. the Moderna (mRNA-1273) or Pfizer-BioNTech (BNT162b2) vaccine are designed to prime neutralizing antibodies against the binding region of the S protein[44].

In summary, we present a detailed immunological study of the SARS-CoV-2-specific T cell response against the E, M and N protein on a single epitope level. PBMC of a well-characterized patient cohort with verified SARS-CoV-2 infection status were examined and stratified into acute and resolved infection. A broadly directed SARS-CoV-2-specific CD4+ T cell response was detectable in 97% of patients regardless of the clinical course. The results of this study add to the body of literature that demonstrates a broad and functional T cell response in most patients with COVID-19. While many peptides elicited a T cell response in one or more patients, there were ten highly recognized peptides that were each recognized by more than a third of patients. These detailed data on SARS-CoV-2-specific T cell epitopes will be helpful for the development of tools like additional SARS-CoV-2-specific MHC class II multimers or to monitor the immune response on an epitope level during future vaccine trials.

## Materials and methods

### Ethics statement

The study was approved by the local ethics board of the Ärztekammer Hamburg (PV4780, PV7298) and written consent was obtained by all study participants.

### Patient cohort

Peripheral blood mononuclear cells (PBMC) from SARS-CoV-2 infected patients (n = 34) and uninfected healthy controls (n = 12) were collected at the University Medical Center Hamburg-Eppendorf. For investigation of *ex vivo* responses an additional eight patients with acute and resolved COVID-19 were recruited (**S1 Table**). Antibody tests confirmed the absence of SARS-CoV-2 antibodies in the uninfected healthy controls. Virus-specific *in vitro* cell cultures were started with fresh, unfrozen PBMC. SARS-CoV-2 infection was verified by at least one positive reverse transcription polymerase chain reaction (RT-PCR) of nasopharyngeal swab in all patients as previously described [18]. Patient disease status was defined as recovered if the first positive PCR was more than 40 days ago, the most recent PCR was negative, and symptoms were resolved. Acute infection was defined as acute symptoms compatible with COVID-19 and/or first positive PCR less than 16 days ago. For patients with acute infections and those with resolved infections, the disease severity, time since onset of symptoms, comorbidities, and clinical outcome were assessed. For patients with acute infection additional clinical parameters including white blood cell and lymphocyte count, c-reactive protein (CRP), hemoglobin, platelet count, further T cell sub-classification as well as comorbidities were assessed at the day of blood sampling for this study (+/- two days) (**Tables 1** and **S1**).

### E, M, and N protein peptides

15-mer peptides overlapping by ten amino acids corresponding to the complete E, M and N protein amino sequences present in the first patient diagnosed in Hamburg [47] were synthesized (peptides & elephants, Hennigsdorf, Germany). 43 peptides cover the M protein, 82 peptides the N protein and 10 peptides the smaller E protein. The synthesis for the peptides Env_P2, Env_P4, and Env_P5 failed, we therefore only tested 10 overlapping peptides spanning the E protein. All peptides were formulated into 13 pools of either 10 or 11 peptides (**S2 Table**). For *in vitro* culture peptide pools were used at a concentration of 1 μg/ml per single peptide. For the enzyme linked immunospot assays (ELISpot) the final concentration of each single peptide was 10 μg/ml. Further peptides for truncation experiments and variants were also synthesized.

### HLA typing

High definition molecular HLA class I and II typing was available for 33 of 34 patients and was performed at the Institute of Transfusion Medicine at the University Medical Center Hamburg-Eppendorf, by PCR-sequence specific oligonucleotide (PCR-SSO) using the commercial kit SSO LabType as previously described (One Lambda, Canoga Park, CA, USA)[48].

### Bulk stimulation of peripheral blood mononuclear cells (PBMC)

30–50 x 10^6 fresh PBMC were divided in 13 wells with 1500 μl of R10 medium (RPMI 1640 medium with 10% FCS (Sigma Aldrich), 1% HEPES buffer and 1% Penicillin-Streptomycin). PBMC of each well were stimulated with one of the 13 peptide pools (**S2 Table**) at a final concentration of 10 μg/ml, together with 1 μg/ml of anti-CD28 and anti-CD49d antibodies (BD FastImmune™, clone: L293 (CD28), clone: L25 (CD49d)) for 10 days. Medium with

recombinant IL-2 (50 U/ml) was added when necessary. After 10 days, cells were re-stimulated with single SARS-CoV-2-peptides (final concentration of 10 μg/ml) and then assayed for interferon-γ (IFN-γ) production by ELISpot and intracellular cytokine staining (ICS) on day 11 as previously described [20].

## ELISpot assay

ELISpot assays were performed as previously described [20, 22, 49]. Cultivated PBMC were stimulated with the respective 10 or 11 peptides from the peptide pool. We used 30.000 cells per well and responses were considered positive if the number of spots was at least three times the number of spots in the negative control and at least a total number of 30 spots. Single peptides were used at a concentration of 10 μg/ml. Anti-CD3-antibodies served as a positive control, R10 + DMSO as a negative control [50]. All positive responses were confirmed by ICS assays following stimulation with the respective peptide.

## Intracellular cytokine staining and flow cytometry

ICS was performed as previously described [20, 49]. 5 x 10^5 PBMC were stimulated with the corresponding SARS-CoV-2 peptide at a final concentration of 10 μg/ml before blocking the secretion with 5 μg/ml Brefeldin A (Sigma Aldrich) one hour after stimulation. Cells were then incubated at 37˚C overnight and stained with the Zombie NIR Fixable Viability kit for live cells as well as surface antibodies against anti-CD3 (clone: Okt3; AlexaFluor 700), anti-CD4 (clone: SK3; PerCP-Cy5.5) and anti-CD8 (clone: RPA-T8; Brilliant Violet 786) (all antibodies by BioLegend). After fixation and permeabilization (eBioscience, Foxp3/Transcription Factor Staining Buffer Set), cells were stained with anti-IFN-γ-antibodies (clone: 4S.B3; PE-Texas red; BioLegend). Cells were then analyzed on a BD LSRFortessa (BD Biosciences). We defined a T cell response as positive when the percentage of CD4+ T cells within the gate for IFN-γ was three times higher than the negative control, above 0.02%, and if the population could be clearly separated from the negative control [20, 49, 51]. R10 and DMSO were added to the negative control. **Fig 1A** shows an exemplary ICS result of an M-specific CD4+ T cell response.

For *ex vivo* ICS, PBMC from COVID-19 patients were stimulated overnight with a peptide pool of the ten most frequently detected N, M and E-specific peptides (Mem_P30, Mem_P36, Ncl_P17, Ncl_P18, Ncl_P70, Ncl_P26, Ncl_P11, Ncl_P71, Ncl_P44, and Env_P12) at a final concentration of 10 μg/ml before blocking the secretion with 5 μg/ml Brefeldin A (Sigma Aldrich) one hour after stimulation. Cells were then stained for IFN-γ (clone: 4S.B3; PE-Texas red; BioLegend), TNFα (clone: Mab11; BV605; BioLegend) and IL-2 (clone: MQ1-17H12; BUV737; BD Biosciences) as described above. The threshold for positivity for the cytokines IFN-γ, TNF-a and IL-2 was set at 0.02% of all CD4+ T cells. The antibody panel used for this assay is shown in **S9 Table**.

## HLA restriction and epitope fine mapping

For epitope fine mapping experiments, PBMC were stimulated with SARS-CoV-2 peptides and corresponding truncated peptides (final concentration of 10 μg/ml) in the presence of IL-2. After 12 days, the cells were re-stimulated with the single peptides and truncations (final concentration of 10 μg/ml) and stained for IFN-γ (ICS). For restriction experiments, the cells were restimulated after T cell culture with PBMC (APC) from healthy, seronegative donors loaded with the respective peptide. The healthy donors matched one HLA-DRB1 molecule of the COVID-19 patients. PBMC from the healthy donors were incubated with the respective peptide (concentration of 10 μg/ml) for 20 minutes and then washed six times. After re-stimulation in the presence of 1:10 donor PBMC, ICS was performed to assess the IFN-γ production

as described above. PBMC from the matched donors served as a negative control, separately peptide-loaded autologous PBMC from the respective patients served as a positive control.

*In vitro* binding assays with 14 of the peptides that elicited a N, M or E-specific CD4+ T cell response were performed using purified HLA-DR molecules, as previously described [52].

### Tetramer staining

The MHC class II tetramer used in this study is specific for a 20-mer peptide sequence (aa145-164 LRGHLRIAGHHLGRCDIKDL) from the SARS-CoV-2 membrane protein restricted by the MHC class II molecule DRB1*11:01. MHC class II tetramer enrichment was performed as previously described [53]. In short, cryopreserved PBMC were thawed and stained with the PE-labelled MHC class II tetramer. Tetramer enrichment was performed using MACS technology with anti-PE microbeads (Miltenyi Biotec, Germany) according to the manufacturer's protocol. Pre-, enriched, and depleted tetramer fractions were further analyzed by flow cytometry using the BD LSRFortessa. The antibody panel can be found in **S9 Table**.

### Antibody screening

IgG antibodies specific to the RBD, the N protein as well as S1 of SARS-CoV-2 and the N protein of the endemic coronaviruses 229E, NL63, OC43 and HKU1 were measured utilizing the commercially available recomLine SARS-CoV-2 IgG (Mikrogen Diagnostik, Neuried, Germany) according to the manufacturer's instructions. In short, 30 samples of acute and resolved COVID-19 patients and four healthy donors were incubated with recombinant antigens of SARS-CoV-2 and endemic coronaviruses. After a washing step, anti-human IgG conjugate antibodies coupled to horseradish peroxidase were added and subsequently unbound conjugate antibodies were washed away. The color reaction catalyzed by the peroxidase was then evaluated in a semiquantitative manner (ranging from "-", indicating no reaction, to "+++", indicating a "very strong intensity"). The whole process from dilution of patient samples to scanning of results was carried out on a CarL "complete automation of recomLine strip assays" device (Mikrogen Diagnostik, Neuried, Germany).

### Statistical analysis

All flow cytometric data were analyzed using FlowJo 10.5.0 software (Treestar, Ashland, OR, USA). Statistical analyses were carried out using the Prism 7.0 software (GraphPad software, San Diego, CA). Mann Whitney test or the Kruskal-Wallis test with Dunn´s post-test was performed throughout all samples for inter-group comparisons. Spearman´s correlation was performed for bivariate correlation analyses. Data are expressed as means with standard deviations (SD) or with standard error of mean (SEM). P-values less than or equal to 0.05 were considered significant.

## Supporting information

**S1 Table. Clinical and immunological patient characteristics.**
(XLSX)

**S2 Table. SARS-CoV-2 peptide sequences of the envelope-, membrane-, and nucleoprotein.**
(XLSX)

**S3 Table. Overview of SARS-CoV-2 envelope protein-specific CD4+ T cell responses in patients with acute and resolved COVID-19.** For each patient the HLA class I and II molecules are listed. aCov = acute COVID-19 patient, rCov = resolved COVID-19 patient. * =

patient was treated with immunosuppressant medication or received chemotherapy.
(XLSX)

**S4 Table. SARS-CoV-2 envelope, membrane and nucleoprotein-specific CD4+ and CD8
+ T cell responses in 12 healthy donors.**
(XLSX)

**S5 Table. Overview of SARS-CoV-2 membrane and nucleoprotein-specific CD8+ T cell
responses in acute and resolved COVID-19 patients.** For each patient, the HLA class I mole-
cules are listed. aCov = acute COVID-19 patient, rCov = resolved COVID-19 patient. * =
patient was treated with immunosuppressant medication or received chemotherapy.
(XLSX)

**S6 Table. Sequence homology of CCC and the most frequently detected SARS-CoV-2 epi-
topes in this study.**
(XLSX)

**S7 Table. Antibody prevalence to CCC as well as SARS-CoV-2.**
(XLSX)

**S8 Table.** (A) In vitro binding value and number of responding patients in this cohort. (B)
Most likely binding HLA-DRB1 molecules for SARS-CoV-2 peptides that elicited a CD4+ T
cell response.
(XLSX)

**S9 Table.** Flow cytometry panels used for *ex vivo* ICS (A) and tetramer analysis (B).
(XLSX)

**S1 Fig. *Ex vivo* ELISpot of two recovered and one acute COVID-19 patient. (A)** *Ex vivo* ELI-
Spot with PBMCs from two resolved and one acute COVID-19 patient stimulated with all 43
membrane peptides. The spots in the negative control ranged from 0 to 1 spot per 100.000
cells. **(B)** *Ex vivo* ELISpot with PBMCs from two resolved and one acute COVID-19 patients
stimulated with all 82 nucleoprotein peptides. The spots in the negative control ranged from 0
to 1 spot per 100.000 cells.
(TIF)

**S2 Fig. Breadth of the CD4+ T cell response against the SARS-CoV-2 envelope, membrane,
and nucleoprotein of patient rCov-17.** All IFN-γ+ CD4+ T cell responses of patient rCov-17
against the envelope, membrane, and nucleoprotein. Gated on CD4+ T cells. All cytokine
gates are set based on the respective negative control (R10 and DMSO).
(TIF)

**S3 Fig.** Breadth of the CD4+ T cell response in correlation with the time after infection **(A)**,
disease severity **(B)**, seroprevalence of IgG antibodies against SARS-CoV-2 **(C)** or CCC **(D)**
and gender **(E)**.
(TIF)

**S4 Fig. Original FACS-plots of restriction assay.**
(TIF)

**S5 Fig. *Ex vivo* ICS with PBMCs of six acute COVID-19 patients after stimulation with the
ten most frequently detected peptides of the envelope, membrane, and nucleoprotein
from this study. (A)** Exemplary IFN-γ, TNF-α and IL-2 CD4+ T cell response, pre-gated on
CD4+ T cells. **(B)** Frequencies of IFN-γ, TNF-α and IL-2 of CD4+ T cells and the distribution

of the frequencies of single, double, and triple positive CD4+ T cells. The threshold for positivity for the cytokines IFN-y, TNF-a and IL-2 was set at 0.02% of all CD4+ T cells. The background detected in the negative controls (R10 + DMSO) was subtracted from positive values. (TIF)

**S6 Fig.** Overview of most frequently detected epitopes in this study and mutations in variants of concern of the **(A)** nucleoprotein, **(B)** membrane protein and **(C)** envelope protein. Horizontal, black parentheses indicate epitopes identified in this study. Red shade indicates asymptomatic CD4+ T cell epitopes identified by Prakash et al. [28] with high conservancy among human and animal coronaviruses. Highlighted amino acids indicate mutations found in variants of concern (according to PANGO lineages: https://cov-lineages.org/). Colour code: yellow, B.1.1.7; light blue, B.1.351; grey, P1 (no VOC-defining mutations are located in the M protein).
(TIF)

## Acknowledgments

We thank all patients who participated in this study. We also thank all of the members of the UKE ID COVID-19 study group for helping with the recruitment of the patients, Silke Kummer, Robin Woost and Melanie Wittner for technical assistance, and Christin Ackermann for the administrative support.

## Author Contributions

**Conceptualization:** Janna Heide, Julian Schulze zur Wiesch.

**Data curation:** Janna Heide, Sophia Schulte, Matin Kohsar, Thomas Theo Brehm, Hendrik Karsten, Julian Schulze zur Wiesch.

**Formal analysis:** Janna Heide, Julian Schulze zur Wiesch.

**Funding acquisition:** Julian Schulze zur Wiesch.

**Investigation:** Janna Heide, Sophia Schulte, Julian Schulze zur Wiesch.

**Methodology:** Janna Heide, Sophia Schulte, Marissa Herrmann.

**Project administration:** Julian Schulze zur Wiesch.

**Resources:** Matthias Marget, Sven Peine, Alexandra M. Johansson, Marc Lütgehetmann, William W. Kwok, Julian Schulze zur Wiesch.

**Supervision:** Julian Schulze zur Wiesch.

**Validation:** Alessandro Sette, John Sidney.

**Visualization:** Janna Heide, Sophia Schulte.

**Writing – original draft:** Janna Heide, Alessandro Sette, John Sidney, Julian Schulze zur Wiesch.

**Writing – review & editing:** Janna Heide, Alessandro Sette, John Sidney, Julian Schulze zur Wiesch.

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
