## [Decision Letter · Decision Letter 0]

11 Jun 2021

Dear Dr. Schulze zur Wiesch,

Thank you very much for submitting your manuscript "Broadly directed SARS-CoV-2-specific CD4+ T cell response includes frequently detected peptide specificities within the membrane and nucleoprotein in patients with acute and resolved COVID-19" for consideration at PLOS Pathogens. As with all papers reviewed by the journal, your manuscript was reviewed by members of the editorial board and by several independent reviewers. In light of the reviews (below this email), we would like to invite the resubmission of a significantly-revised version that takes into account the reviewers' comments.

Please address the many points of clarification raised by the reviewers. In particular, both reviewers requested that the authors contextualize their findings regarding SARS-CoV-2-specific CD4+ T-cell epitopes with possibly conserved epitopes found in common cold coronaviruses. Furthermore, both reviewers were unconvinced that the data support the Th1-biased responses the authors claim. Please provide additional evidence to substantiate these claims as suggested by the reviewers, or tone down such claims. If additional data supports the Th1-biased response claimed by the authors, then supplementary Fig.3 should be moved to the main text. Finally, for additional clarity and transparency, the authors should consider moving the exemplar data in Supplementary Fig. 2 to the main text. This should be considered along with Reviewer #1's request to show the original FACS plots for the HLA-restriction experiments in Fig. 3B. 

We cannot make any decision about publication until we have seen the revised manuscript and your response to the reviewers' comments. Depending on the comprehensiveness and clarity of your point-by-point response, I will decide whether your revised manuscript needs to be sent back to reviewers for further evaluation.

Sincerely,

Benhur Lee

Section Editor

PLOS Pathogens

Benhur Lee

Section Editor

PLOS Pathogens

Kasturi Haldar

Editor-in-Chief

PLOS Pathogens

orcid.org/0000-0001-5065-158X

Michael Malim

Editor-in-Chief

PLOS Pathogens

orcid.org/0000-0002-7699-2064

Reviewer's Responses to Questions

**Part I - Summary**

Reviewer #1: In the manuscript entitled “Broadly directed SARS-CoV-2-specific CD4+ T cell response includes frequently detected peptide specificities within the membrane and nucleoprotein in patients with acute and resolved COVID-19” by Heide et al., the authors analyze T cell responses (mostly CD4, but also some information on CD8) against the envelope (E), membrane (M) and nucleoprotein (N) in a cohort of 34 patients from Hamburg by performing in vitro stimulation with overlapping peptides. They detect few responses targeting the E-protein, but several responses targeting the M- and N-proteins. The responses against M- and N-derived peptides are almost evenly spread across the proteins and several peptides are targeted by almost all patients in this cohort which lets the authors to conclcude that these epitopes can be presented by a number of different HLA-alleles.

Collectively, the authors present an interesting piece of work that certainly adds to the literature regarding SARS-CoV-2-specific immunity. While the study is well performed and the paper is nicely written, I have several comments the authors might want to consider to improve the overall strength of their manuscript.

Reviewer #2: In this study by Heide et al, they characterised the breadth and specificity of the SARS-CoV-2 peptide specific response to structural proteins E, M and N in a cohort of 10 acute and 24 convalescent volunteers. They use tried and trusted techniques like the short-term cultured ELISpot to show T cell immunodominant responses at the population (response frequency) and individual (mapped responses) levels. They go further to describe the HLA allotypes restricting these responses and then finally phenotype these responses using ICS. Here, they encounter a challenge as most studies have found in that the detection of SARS-CoV-2 specific responses using ICS is challenging due to the low sensitivity of ICS and the comparatively lower frequencies of circulating SARS-CoV-2 cells in the periphery compared to virus-specific circulating frequencies observed in chronic infections. Their epitope mapping in a group of healthy controls will also help to define source of the cross-reactive epitopes reported in some of the literature.

In summary, their results contribute to the growing body of work identifying SARS-CoV-2 epitopes that could prove to be important addressing cross-reactive epitopes from other infections and in the design of the next generation SARS-CoV-2 vaccines.

**Part II – Major Issues: Key Experiments Required for Acceptance**

Reviewer #1: Specific points:

The manuscript lacks information on common cold coronaviruses (CCC) and the potential of pre-existing immunity against the analyzed proteins from these CCCs. Could the authors provide information on how closely related the fine-mapped and most targeted epitopes are to the corresponding regions in CCCs?

I am also unsure about the precise message the authors are trying to convey with regards to HLA-restriction. This pertains mostly to the HLA-binding information the authors provide. Some peptides are targeted by almost all patients, irrespective of the patients’ class II HLA alleles. In addition, some patients target several consecutive peptides beyond the mere overlapping regions between two peptides. The authors also discuss these highly promiscuous HLA class II binders, however, the HLA fine mapping experiments are rather limited in this context. Are these peptides indeed presented by so many HLA molecules?

Are the data shown in figure 1B derived ex vivo or after stimulation?

Were responses in patients (especially resolved patients) stratified by the time post infection? It would be important to see whether there is a correlation. Also, all acute patients were displayed together despite differences in clinical courses (22 mild and 12 severe including 7 with ICU treatment). Did this affect T cell responses? Were they sampled early? Have they already received ICU treatment/steroids?

In Figure 3B, it would be helpful to see the original FACS plots from the HLA-restriction experiments.

The authors discuss that “we detected no significant difference between the number and magnitude of T cell responses in early acute vs. resolved infection” (lines 267f), however, in table 2, it appears as if aCov patients had fewer responses against membrane compared to rCov patients (at least in the responses between peptides 5 and 25). Can this be quantified?

Why were the additional patients recruited “for investigation of ex vivo responses” (line 341) not listed in table 1? Were the uninfected healthy controls historic controls prior to the beginning of the pandemic?

I am not convinced that we are truly looking at a Th1 bias in these cells, the provided data are too limited in my view to draw this conclusion.

The variants of concern should be classified following the new WHO label.

Reviewer #2: (No Response)

**Part III – Minor Issues: Editorial and Data Presentation Modifications**

Reviewer #1: (No Response)

Reviewer #2: 1. The manuscript was easy to read and well-written however, the figures could be improved with more granularity in terms of information e.g. immunosuppressed patients and acute vs convalescent subject in the cohort could be distinguished using different colours or shape for in-depth information in figure 1B.

2. Line 55 – needs an update

3. Line 57 – can authors explain what they mean by ‘general fragility’ as a risk factor for severe disease and why co-morbidities does not suffice

4. Consider using convalescence instead of reconvalescence.

5. Table 1 – what do the range under blood count, immunology and clinical parameters at the time of analysis represent? Are these the range in healthy controls or range in their cohort. Also, why is there none of this information for the resolved COVID-19 cohort?

6. Consider changing label for Y axis on Figure 1B for clarity. e.g. number of CD4+ responses in each subject.

7. Why were 4 patients excluded in Figure 1B?

8. Add reference to supplementary table 1 in text for summary data in lines 157-160

9. Figure 1A and Supplementary figure 2 and 3A - What are cytokine gates set based on? Unstimulated (negative) controls or FMO? If negative controls, did they use the R1O or DMSO control? DMSO control would be the preferred control for setting gates here. This information should be clear in the figure legends. Gating in supplementary figure 2 seems to be quite stringent to pick up strong responders, this makes for robust data however the rationale for this should be clearly explained in text. Responses like NCL-P11 amongst others appear low to be a ‘true positive’. How did the authors confirm these extremely low responses? Concerns would be around this inflating the response frequency reported in the manuscript.

10. Supplementary figure 3A – what did the full T cell flow cytometer panel look like. A supplementary table will be panel will be useful. Authors report that the responses are Th1 however, it seems the panel was set up to only detect Th1-biased responses (Ifng, Tnf, IL-2) in the first place. Were other cytokines used and not detected? Perhaps clarify that they had sought to describe Th1 responses as the assay was not set up to detect any other Th-biased response other that Th1. While I acknowledge that non-Th1 cytokines can be difficult to assay ex vivo with standard ICS and may have informed the panel design and result presentation, author’s wordings suggest that other Th subsets were evaluated and not found. If other Th phenotypes are desired, then a cytokine independent approach e.g., AIM assay would be the way to go. The AIM assay is also more sensitive than the ICS and would be useful in quantifying the magnitude of these responses.

11. Flow cytometry – threshold for positivity for Ifng+, Tnf+, and IL-2+ should be mentioned in main text.

12. Line 268 which addresses evolution of response at a population level could be made into a summary table

13. Line 282 – update to new WHO names for variants of concern (VOC)

14. Are there any experiments evaluating the change, if any, of the T cell responses when peptides covering mutated regions of VOCs are used to re-stimulate short term cell lines?

15. Line 315 – 317 – the authors need to be careful about this statement. I do not believe their data has the power to confirm this so it is speculative and may be stretching their conclusions.

16. Typo in table 1 – Days sinc(y)e start of symptoms

17. Supplementary table 1 - Consider ordering this based on clinical phenotypes. Perhaps add nationality to the table

18. Did country of origin play a role in study findings e.g. on the breadth of responses and do T cell responses targeting particular epitopes impact on recovery or associate with VL?

19. Did authors check for antibody or T cell responses to CCC in this cohort?

20. Line 321 – Can the authors comment on the incidence of re-infection after natural infection vs after vaccination given the breadth of the responses following natural infection?

21. The healthy control data seems to stand alone. Can they integrate this into their data e.g were there shared epitopes? Are shared responses targeting conserved regions? Were they expanded by infection?

PLOS authors have the option to publish the peer review history of their article (what does this mean?). If published, this will include your full peer review and any attached files.

Reviewer #1: No

Reviewer #2: No
---

## [Editor Report · Decision Letter 1]

27 Jul 2021

Dear Prof. Schulze zur Wiesch,

We are pleased to inform you that your manuscript 'Broadly directed SARS-CoV-2-specific CD4+ T cell response includes frequently detected peptide specificities within the membrane and nucleoprotein in patients with acute and resolved COVID-19' has been provisionally accepted for publication in PLOS Pathogens.

Best regards,

Benhur Lee

Section Editor

PLOS Pathogens

Benhur Lee

Section Editor

PLOS Pathogens

Kasturi Haldar

Editor-in-Chief

PLOS Pathogens

orcid.org/0000-0001-5065-158X

Michael Malim

Editor-in-Chief

PLOS Pathogens

orcid.org/0000-0002-7699-2064
---

## [Editor Report · Acceptance letter]

23 Aug 2021

Dear Prof. Schulze zur Wiesch,

We are delighted to inform you that your manuscript, "Broadly directed SARS-CoV-2-specific CD4+ T cell response includes frequently detected peptide specificities within the membrane and nucleoprotein in patients with acute and resolved COVID-19," has been formally accepted for publication in PLOS Pathogens.

Best regards,

Kasturi Haldar

Editor-in-Chief

PLOS Pathogens

orcid.org/0000-0001-5065-158X

Michael Malim

Editor-in-Chief

PLOS Pathogens

orcid.org/0000-0002-7699-2064